# Stress and Associated Factors among Frontline Healthcare Workers in the COVID-19 Epicenter of Da Nang City, Vietnam

**DOI:** 10.3390/ijerph18147378

**Published:** 2021-07-10

**Authors:** Nhan Phuc Thanh Nguyen, Duong Dinh Le, Robert Colebunders, Joseph Nelson Siewe Fodjo, Trung Dinh Tran, Thang Van Vo

**Affiliations:** 1Institute for Community Health Research, University of Medicine and Pharmacy, Hue University, Hue 530000, Vietnam; nptnhan@huemed-univ.edu.vn; 2Faculty of Public Health, University of Medicine and Pharmacy, Hue University, Hue 530000, Vietnam; ldduong@hueuni.edu.vn; 3Global Health Institute, University of Antwerp, 2610 Antwerp, Belgium; robert.colebunders@uantwerpen.be (R.C.); JosephNelson.SieweFodjo@uantwerpen.be (J.N.S.F.); 4Faculty of Public Health, Da Nang University of Medical Technology and Pharmacy, Da Nang 550000, Vietnam; trandinhtrung@dhktyduocdn.edu.vn

**Keywords:** COVID-19, stress, Vietnam, healthcare workers

## Abstract

Frontline healthcare workers (HCWs) involved in the COVID-19 response have a higher risk of experiencing psychosocial distress amidst the pandemic. Between July and September 2020, a second wave of the COVID-19 pandemic appeared in Vietnam with Da Nang city being the epicenter. During the outbreak, HCWs were quarantined within the health facilities in a bid to limit the spread of COVID-19 to their respective communities. Using the stress component of the 21-item Depression, Anxiety and Stress Scale (DASS-21), we assessed the level of stress among HCWs in Da Nang city. Between 30 August and 15 September 2020, 746 frontline HCWs were recruited to fill in an online structured questionnaire. Overall, 44.6% of participants experienced increased stress and 18.9% severe or extremely severe stress. In multivariable analysis, increased stress was associated with longer working hours (OR = 1.012; 95% CI: 1.004–1.019), working in health facilities providing COVID-19 treatment (OR = 1.58, 95% CI: 1.04–2.39), having direct contact with patients or their bio-samples (physicians, nurses and laboratory workers; OR = 1.42, 95% CI: 1.02–1.99), low confidence in the available personal protective equipment (OR = 0.846; 95% CI: 0.744–0.962) and low knowledge on COVID-19 prevention and treatment (OR = 0.853; 95% CI: 0.739–0.986). In conclusion, many frontline HCWs experienced increased stress during the COVID-19 outbreak in Da Nang city. Reducing working time, providing essential personal protective equipment and enhancing the knowledge on COVID-19 will help to reduce this stress. Moreover, extra support is needed for HCWs who are directly exposed to COVID-19 patients.

## 1. Introduction

In December 2019, an outbreak of the novel coronavirus 2019 disease (COVID-19) was declared in Wuhan, China and rapidly spread to other countries. By March 2020, COVID-19 had evolved into a pandemic that subjected healthcare systems and workers to tremendous pressure [1]. Within a short period of time, COVID-19 became a global crisis and significantly impacted all aspects of life. Indeed, as of 2 May 2021, more than 152 million confirmed COVID-19 cases and 3.2 million related deaths had been reported.

The first case of COVID-19 was reported in Vietnam on 23 January 2020, and the Vietnamese government urgently rolled out emergency policies across the entire country. During the first wave of the outbreak, almost all infected cases were imported and quarantined, with no deaths reported. Although Vietnam has experienced many infectious disease outbreaks in the past, COVID-19, as a new entity, constituted a crucial challenge for the local healthcare system and healthcare workers (HCWs). After 99 days without any community cases, a second wave of the COVID-19 pandemic appeared in Vietnam on 17 July 2020 with an epicenter in Da Nang city, a tourist city in the central coastal region. From 17 July to 10 September 2020, a total of 551 cases were reported countrywide. Most of these cases were related to a history of a recent visit to Da Nang. During the second COVID-19 wave in Vietnam, the Da Nang general hospital became an infection hotspot with at least 246 COVID-19 cases reported among inpatients, caregivers and HCWs (19 cases). Moreover, secondary infections spreading from the hospital to the community were observed [2]. This outbreak resulted in an increased workload and prolonged working time for frontline HCWs. Adding to the overwhelming work burden, many HCWs were also quarantined within health facilities together with COVID-19 patients and care givers in a bid to limit the spread of the COVID-19 to their respective communities [3]. HCWs were kept away from their families and children for an average of 30 days but food supplies and necessities were provided to them by the government while they were in quarantine.

HCWs constitute a high-risk group during the ongoing COVID-19 pandemic. Indeed, high morbidity and mortality due to COVID-19 have been reported among HCWs across the globe. In May 2020, it was estimated that about 150,000 HCWs had contracted COVID-19, with an estimated 1400 deaths worldwide [4]. Since then, these numbers have been increasing rapidly. By 7 April 2021, only in the United States, more than 3600 HCW deaths were reported, and the majority of them died under the age of 60 (mean age 59) [5]. Confronted with the COVID-19 pandemic, HCWs are exposed to both physical and psychological stressors that affect their mental health. Factors such as insufficient personal protective equipment (PPE), long working hours, heavy workload, and anxiety about getting sick and possibly infecting loved ones would likely compound the stress experienced by HCWs involved in the COVID-19 response. In addition, stigma against HCWs working in high-risk environments can have a profound effect on their mental health [6]. Several studies have shown a high prevalence of stress among HCWs, ranging from 2.2% to 41.2% depending on their specialization, type of activity and proximity to COVID-19 patients [7,8,9]. Prolonged stress not only will have a negative effect on the physical and mental health of HCWs, but also affect their work performance [10].

In this study, we investigated the stress level and factors associated with stress among frontline HCWs who were involved in the COVID-19 response in the epicenter of Da Nang in Vietnam. In addition, this study aimed to provide evidence for policymakers to implement effective interventions to reduce stress and increase resilience among frontline HCWs during the COVID-19 pandemic.

## 2. Materials and Methods

### 2.1. Study Setting and Design

This was a cross-sectional online survey conducted from 30 August to 15 September 2020 in Da Nang, Vietnam. This was the 3rd online survey initiated by the ICPCovid consortium (https://www.icpcovid.com/ accessed on 14 July 2020) to assess the impact of the COVID-19 pandemic in Vietnam.

### 2.2. Participants

Frontline HCWs working in the public healthcare system and private hospitals in Da Nang city during the pandemic period were asked to participate in an online survey. A frontline worker was defined as a healthcare staff member who was actively involved in the COVID-19 response (diagnosis, treatment, prevention or public health activities) during the outbreak in Da Nang city. An official invitation letter to participate in the survey was sent by the Da Nang health department to all medical facilities, including community health centers, and the Da Nang general hospital. It is estimated that there are about 10,000 HCWs in Da Nang city. HCWs received this letter through their local administrative staff. Those who consented to participate then accessed the online survey tool to submit their responses. Given the sampling approach which we adopted, we opted for a convenient sample size whereby all eligible responses received within the study period would be analyzed.

### 2.3. Data Collection

A Google survey form was created using a structured questionnaire. The survey link was sent to the administrative departments of all health facilities involved in the COVID-19 response in Da Nang city. This questionnaire was based on a questionnaire developed by the ICPCovid consortium but was adapted to the COVID-19 situation in Vietnam. Data were collected anonymously. Consenting participants completed the questionnaire and provided the following data:

Socio-demographic characteristics: Age, gender, marital status, professional qualifications, years of working experience, healthcare facility where they practiced, whether they lived with a vulnerable person (i.e., children < 12 years; elderly persons; chronically ill persons).Working conditions: Total working time per week (in hours); night shift situation (yes/no). Self-perceived knowledge about infection control specific to COVID-19; self-perceived confidence in available personal protective equipment (PPE) such as masks, gloves, hand sanitizers, protective clothing for COVID-19 prevention at the workplace. Self-perceived data were collected using a 10-point Likert scale (1 = minimal level, to 10 = maximal level).Stress level: The stress component (7 questions) of the 21-item Depression, Anxiety and Stress Scale (DASS-21) was used to evaluate the stress status. Scores were dichotomized into normal stress (with scores between 0 and 14) and increased stress (with scores greater than 14). Increased stress was further classified into mild (with scores between 15 and 18), moderate (with scores between 19 and 25), severe (with scores between 26 and 33) and extremely severe stress (with scores greater than 33) [11].Self-perceived support for HCWs during the COVID-19 pandemic: A 10-point Likert scale (1 = not having support, to 10 = enthusiastic support) was used to measure the perceived support enjoyed by HCWs considering three sources of support: society, co-workers and HCWs’ families and relatives. A higher score indicated a higher level of support.◦Support from society: HCWs’ perception of the society’s support towards them, in the form of material support (such as provision of protective equipment) and non-material support (such as words of appreciation and encouragement through mail, mass media, social networks).◦Support from co-workers: HCWs’ perceptions of support, such as sharing of workloads, and mutual encouragement of colleagues.◦Support from family or relatives: HCWs’ perceptions of receiving encouragement, material support (such as food, bottled water, toiletries) and non-material support (such as phone calls, prayers) from family and relatives.

### 2.4. Statistical Analysis

All statistical analyses were performed using Stata 15.0. Descriptive statistics were used to depict the demographic and occupational characteristics, as well as stress levels of HCWs using numbers with percentages, means with standard deviation or medians with interquartile ranges. A multiple logistic regression model was used to determine factors associated with increased stress among HCWs. Stress outcomes were dichotomized as follows: no stress (coded as 0) and stress (coded as 1). Covariates included demographic variables (age, sex, marital status and living with vulnerable groups in the same home), self-perceived support to HCWs, knowledge about COVID-19 prevention and treatment, confidence in the available PPE, type of profession and type of healthcare facility.

HCW professions were summarized into two groups, depending on whether the profession exposed the HCW to direct contact with patients/bio-samples. Accordingly, physicians, nurses and laboratory workers constituted one group (high contact), while pharmacists, public health officers and others formed the group of “low contact” HCWs. We also compared two groups of healthcare facilities: those that served as COVID-19 treatment units (public hospitals at the city and district level, and private hospitals within the city) and facilities not providing COVID-19 treatment such as the city Center for Disease Control and Prevention (CDC), the emergency transport system, community health centers, contact tracing units, logistic/administrative support units and testing centers. All inferential analysis was considered statistically significant at *p*-value < 0.05.

### 2.5. Ethical Considerations

Anonymity and informed consent were ensured via online registration of the survey. The study was officially permitted by the Health Department of Da Nang city and previously approved by the Ethical Review Committee of Hue University of Medicine and Pharmacy, Vietnam (No. H202/041).

## 3. Results

A total of 1011 HCWs voluntarily completed the online questionnaire but after cleaning the data and applying the inclusion criteria, only 746 (73.8%) HCWs were included in the analysis. For the 746 participants, the mean age was 32.8 ± 8.9 years, 72.5% were women, 64.8% married and 78.3% lived with at least one person from a vulnerable group. A large proportion (43.4%) were nurses, and the median number of years of working experience was 6. Of the participants, 68.8% were either a physician, nurse or laboratory workers and 77.3% worked in COVID-19 treatment units. Nearly half (44.6%) of the HCWs who participated in the study experienced stress (score > 14) during the COVID-19 outbreak in Da Nang city.

Stress distribution in occupational groups that have regular direct contact with patients and bio-samples, such as physician, laboratory worker or nurse, was 50.3%, 50% and 46.3%, respectively (Table 1).

Up to 116 (15.5%) and 25 (3.3%) of participants experienced severe and extremely severe stress, respectively (Table 2). The median stress score was 14 (on a scale ranging from 0 to 42).

The median number of working hours per week was 48 h (IQR: 40–56). Only 12.2% of HCWs reported night shifts during the outbreak period. An increasing number of working hours per week increased the odds of stress by 12‰ (OR = 1.012; 95% CI: 1.004–1.019) (Table 3). The odds of stress varied inversely with the level of confidence in the available personal protective equipment (OR = 0.846; 95% CI: 0.744–0.962) (Table 3). An increased score of HCWs’ knowledge of COVID-19 reduced the odds of experiencing stress (OR = 0.853; 95% CI: 0.739–0.986). Physicians, nurses and laboratory staff (high-contact HCWs) had higher odds of stress (OR = 1.42 (95% CI: 1.02–1.99). Similarly, HCWs working in treatment units showed significantly higher odds of experiencing stress (OR = 1.58, 95% CI: 1.04–2.39).

## 4. Discussion

Our survey provides insights into the stress experienced by frontline HCWs in Vietnam, a country that has been quite successful in controlling the COVID-19 pandemic. We investigated the working conditions of HCWs and factors associated with stress during a COVID-19 outbreak in Da Nang city. Of the 746 HCWs who participated in the study, 44.6% experienced stress. Increasing working time, confidence in the available PPE, knowledge about COVID-19 prevention and treatment, work in healthcare facilities providing COVID-19 treatment, and being a physician, nurse or a laboratory worker were independent predictors of stress. A particular aspect of the way the COVID-19 outbreak was managed in Da Nang and that may have increased the stress among HCWs was the long quarantine period during which certain HCW were separated from their families to prevent further community transmission.

Another survey conducted among HCWs in Vietnam shortly after the first COVID-19 wave (end of April 2020) found that 34.3% HCWs experienced stress measured with the Revised Impact of Event Scale (IES-R) tool [12]. In a survey conducted from April–June 2020 in five countries in the Asia-Pacific region, also using the IES-R tool, a stress prevalence of 3.3% was reported among HCWs in Vietnam, but only 50 Vietnamese HCWs participated in this survey [13]. These differences may be explained by the time these studies were conducted. In April–June 2020, the number of COVID-19 cases nationwide was only 381, while during the outbreak in Da Nang, 551 cases were recorded only in one city [2]. Worldwide, it has been reported that an increase in disease burden increases the concern about becoming infected and therefore may lead to increased stress [9,12,14]. In a recent systematic review of 35 papers with data from 25,343 medical staff, a high level of perceived stress was reported by 56% of them (95% CI = 32–79%) [14]. Using the same stress scoring system (DASS-21) as in our study, 23.8% of HCWs in Oman experienced stress [15], and 41.2% in Turkey [9]. A much higher prevalence of stress among HCWs was observed in China, Canada and Pakistan: 71.5%, 85.6% and 90.1%, respectively [16,17,18]. However, the latter countries were confronted with a more severe COVID-19 disease burden compared to Vietnam.

In our study, the prevalence of stress in the group directly in charge of treating and taking care of patients, who were physicians and nurses, was 50.3% and 46.3%, respectively (Table 1). Overload of work, contact with severely ill COVID-19 patients, along with the fear of infection and infecting relatives, are factors that cause stress in this group [19,20]. A high prevalence of stress (50%) was also observed among laboratory workers. In Da Nang, laboratory workers also draw blood or obtain nose or throat swabs. Handling of bio-samples from suspected or infected patients increases the risk of exposure, leading to increased stress [20,21,22]. In addition, too much work and wearing PPE for long periods of time in hot weather exposed laboratory workers to heat stress, affecting both cognitive and physical performance, leading to increased stress [22,23,24].

The outbreak in Da Nang was mainly a nosocomial outbreak that started in the main hospitals. This led to a rapid increase in cases of COVID-19. To ensure patient care and epidemic control in Da Nang, the medical staff had to face a huge workload, often with limited resources. In addition to providing routine health services, HCWs had to undertake additional tasks ranging from contact tracing, monitoring, testing and treating COVID-19 patients. Our findings indicated that pharmacists and public health officers experienced less stress, most likely because they were less likely to have direct contact with patients and they were not quarantined during the outbreak. In addition, we found that the number of working hours per day and the number of working days per week both exceeded the prescribed maximal working duration for employees in Vietnam (48 h/week) (Table 3). These results were consistent with other studies worldwide showing the overwhelming workload among HCWs during the pandemic [25,26]. This increase in daily working hours and number of working days per week increased the risk of stress among HCWs [9,27].

In our study, better knowledge about COVID-19 prevention and treatment was associated with less stress. Therefore, training HCWs to improve the clinical management of persons with COVID-19 disease and training them to protect themselves to decrease the risk of infection will make them more confident in patient care and will reduce stress. Other studies reported that occupational protection practices and training people to increase their COVID-19 occupational protection knowledge reduced stress and prevented psychological problems [28,29].

Similar to other studies, confidence in the available PPE and protective measures reduced the likelihood of stress [8,25]. Indeed, being equipped with quality PPE will help HCWs to feel protected from contracting the virus, and also limit their risk of infecting family members when they eventually return home [30].

Our research showed that HCWs who worked in health facilities treating COVID-19 patients had a higher risk of stress than HCWs working in facilities not providing COVID-19 treatment (Table 3). Close and frequent contact with COVID-19 patients, working longer hours than usual and working in isolation units have been recognized as factors that increase the likelihood of infection, affecting the health and lives of HCWs, and increasing the risk of stress [17,31,32,33]; all these conditions were fulfilled in the COVID-19 treatment centers in Da Nang.

Thanks to the rapid implementation of drastic preventive measures and the efforts of the HCWs, the COVID-19 outbreak in Da Nang was rapidly controlled. More than a fifth of Da Nang city residents (208,028) were tested for COVID-19; 454 people were quarantined in healthcare facilities; 15,120 in centralized quarantine facilities; and 15,079 self/home-quarantined. By 4 September 2020, all lockdown measures were lifted and on 23 September the last COVID-19 patient in Da Nang was discharged from the hospital. However, in May 2021, a new COVID-19 outbreak appeared in Vietnam, mainly in the city of Hanoi but also with community transmission in other cities, including Da Nang. Lockdown measures were reinstituted. By 13 May, only nearly 0.98% of the population in Vietnam had received at least one dose of a COVID-19 vaccine. Therefore, to control this new wave as well as future COVID-19 outbreaks, scaling up the COVID-19 vaccination campaign will be needed.

We acknowledge that our study had several limitations. As data were collected via an online survey, we do not know whether the HCWs who participated in the survey were representative of all HCWs in Da Nang. We were able to analyze only the responses from less than 10% of the estimated 10,000 HCWs in Da Nang city. Moreover, we cannot verify the validity of the answers to the survey questions. Recall bias and social desirability may have affected the quality of data provided by some HCWs. Moreover, several potential stress factors, such as perceived job demand, job control, economic reward and personal reward, were not investigated. Finally, due to some unavoidable limitations in terms of convenient sampling technique and small sample size, in our multiple logistic regression we could not include each occupational group, but we classified them into two categories based on criteria of direct and indirect contact with patients and bio-samples.

## 5. Conclusions

Stress was commonly experienced by frontline HCWs during the COVID-19 outbreak in Da Nang city, Vietnam. Keeping HCWs quarantined in health facilities was most likely an important factor in containing the outbreak in Da Nang. However, this approach may have increased the stress experienced by HCWs. Given the key role frontline HCWs play in fighting the COVID-19 pandemic, it is of great importance to implement strategies to improve their well-being. Reducing working time, providing full PPE and increasing HCWs’ knowledge about COVID-19 prevention and treatment will help to reduce stress and to increase their effectiveness to control the COVID-19 outbreak. Moreover, extra support is needed for HCWs who are directly exposed to COVID-19 patients. HCWs and persons at risk for severe COVID-19 disease should be priority populations for COVID-19 vaccination.

## Figures and Tables

**Table 1 ijerph-18-07378-t001:** Demographic and occupational characteristics of healthcare workers grouped by the reported stress levels (*n* = 746).

*p*	Overall	Experienced Stress	No Stress
Number of participants (%)	746 (100.0%)	333 (44.6%)	413 (55.4%)
Age in years: mean (SD)	32.8 ± 8.9	32.2 ± 8.7	35.0 ± 9.1
Sex: *n* (%)			
Male	205 (27.5%)	96 (46.8%)	109 (53.2%)
Female	541 (72.5%)	237 (43.8%)	304 (56.2%)
Marital status: *n* (%)			
Married	483 (64.8%)	210 (43.5%)	273 (56.5%)
Single or divorced	263 (36.2%)	123 (46.8%)	140 (53.2%)
Living with vulnerable groups: *n* (%)			
Yes	584 (78.3%)	254 (43.5%)	330 (56.5%)
No	162 (21.7%)	79 (48.8%)	83 (51.2%)
Work experience (years): median (IQR) *	6 (2–11)	5 (2–10)	7 (2.5–12)
Profession Physician Nurse Laboratory worker Pharmacist Others **	147 (19.7%) 326 (43.7%) 40 (5.4%) 65 (8.7%) 168 (22.5%)	74 (50.3%) 151 (46.3%) 20 (50.0%) 18 (27.7%) 70 (41.7%)	73 (49.7%) 175 (53.7%) 20 (50.0%) 47 (72.3%) 98 (58.3%)
Healthcare facilities COVID-19 treatment units Not treatment units (city CDC and others **)	577 (77.3%) 169 (22.7%)	271 (47.0%) 62 (36.7%)	306 (53.0%) 107 (63.3%)

* IQR: interquartile range; ** public health officers, emergency transport systems, community health centers, contact tracing unit, logistic/administrative support unit, testing center.

**Table 2 ijerph-18-07378-t002:** Stress levels of the healthcare workers in Da Nang (*n* = 746).

Stress Levels	Number (%)
Normal stress (score ≤ 14)	413 (55.4)
Increased stress (score > 14)	333 (44.6)
Mild (15 ≤ score ≤ 18)	108 (14.5)
Moderate (19 ≤ score ≤ 25)	84 (11.3)
Severe (26 ≤ score ≤ 33)	116 (15.5)
Extremely severe (score > 33)	25 (3.3)
Overall stress scale score: median (IQR)	14 (6–22)

IQR: interquartile range.

**Table 3 ijerph-18-07378-t003:** Predictors of stress among healthcare workers by multiple logistic regression (*n* = 746).

Factors	OR adj *	*p*	95% CI	
Total number of working hours/weeks	1.012	0.002	1.004	1.019
Support from co-workers	1.005	0.943	0.885	1.141
Support from family or relatives	0.987	0.825	0.879	1.108
Support from society	0.916	0.051	0.838	1.001
Confidence in the available personal protective equipment	0.846	0.011	0.744	0.962
Self- reported knowledge of COVID-19 prevention and treatment	0.853	0.031	0.739	0.986
Profession				
Pharmacist, public health officer and others	Ref			
Physician, nurse, laboratory worker	1.42	0.039	1.02	1.99
Healthcare facilities				
Not treatment units (city CDC and others **)	Ref			
COVID-19 treatment units (public hospitals in city and district, private hospitals in city)	1.58	0.032	1.04	2.39

* Odds ratio adjusted by age, sex, marital status and living with vulnerable groups; ** public health officers, emergency transport systems, community health centers, contact tracing unit, logistic/administrative support unit, testing center.

## Data Availability

All responses were anonymous and securely stored in a password-protected computer at the Institute for Community Health Research, Hue University of Medicine and Pharmacy.

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
