# Peer review of "Stress and Associated Factors among Frontline Healthcare Workers in the COVID-19 Epicenter of Da Nang City, Vietnam"

_ijerph, 2021, doi:10.3390/ijerph18147378_

Round 1

Reviewer 1 Report

Dear authors,

You have done research on stress related to frontline healthcare workers. This work was carried out in Vietnam during the COVID-19 pandemic.

The authors must answer some questions:

 INTRODUCTION

The authors have not correctly formulated the objective of the study. You must be more precise.

 MATERIALS AND METHODS

Participants:

How was the sample chosen? The authors must specify it.

The authors must include the response rate of the participants in the study.

Data collection

The authors took into account whether the study subjects worked the night shift. Did you not take the turnover into account?

RESULTS

The authors must give the results distinguishing each professional category. The functions and competencies of a doctor are different from a nurse. The same happens with other professionals.

 DISCUSSION

The "discussion" cannot be assessed until the authors separate their "results" by professional categories.

 REFERENCES

The references do not meet the journal guidelines. The authors have mixed APA and Vancouver regulations.

Some references are incomplete or have errors. The authors should review this section.

Author Response

Dear Reviewer 1,

Thank you very much for your very useful comments to improve our manuscript better as expected. Please see attachment file is our response to your comments 

Reviewer 2 Report

Dear Authors

I carefully evaluated the present paper, finding it overall interesting and well presented. Nevertheless, several concerns have to be solved before considering it as suitable for publication.

Here my comments.

Introduction: the introduction must be improved. It is not clear the scientific background of the study, whether it considers stress related effects or other outcomes such as infection and death of HCW directly involved in COVID19 care.  Moreover, since several scientific papers have already reported stress related effect of COVID-19 in HCW, the literature gap is very questionable. I suggest to strongly improve scientific background focusing on mental health effects of COVID-19. Authors can also refer to these updated articles:

Pollock A, Campbell P, Cheyne J, Cowie J, Davis B, McCallum J, McGill K, Elders A, Hagen S, McClurg D, Torrens C, Maxwell M. Interventions to support the resilience and mental health of frontline health and social care professionals during and after a disease outbreak, epidemic or pandemic: a mixed methods systematic review. Cochrane Database Syst Rev. 2020 Nov 5;11:CD013779. doi: 10.1002/14651858.CD013779.

Carmassi C, Foghi C, Dell'Oste V, Cordone A, Bertelloni CA, Bui E, Dell'Osso L. PTSD symptoms in healthcare workers facing the three coronavirus outbreaks: What can we expect after the COVID-19 pandemic. Psychiatry Res. 2020 Oct;292:113312. doi: 10.1016/j.psychres.2020.

Giorgi G, Lecca LI, Alessio F, Finstad GL, Bondanini G, Lulli LG, Arcangeli G, Mucci N. COVID-19-Related Mental Health Effects in the Workplace: A Narrative Review. Int J Environ Res Public Health. 2020 Oct 27;17(21):7857. doi: 10.3390/ijerph17217857

Preti E, Di Mattei V, Perego G, Ferrari F, Mazzetti M, Taranto P, Di Pierro R, Madeddu F, Calati R. The Psychological Impact of Epidemic and Pandemic Outbreaks on Healthcare Workers: Rapid Review of the Evidence. Curr Psychiatry Rep. 2020 Jul 10;22(8):43. doi: 10.1007/s11920-020-01166-z

Moreover, study motivation, study question and study aims must be clearly reported.

Materials and Methods: are the questionnaires validates in your language? Authors should motivate the reason why they used a classification of stress, instead of the scores. The classification into several categories such as mild or high stress is very questionable. Stress in the workplace is a complex effect of several determinants that should be properly evaluated.

Results: this section is lacking. Several variables you have collected are now simply omitted. For example, what is the result of three way of support? Nothing to say about other variables such as perceived job demand, job control, economic reward, personal reward and so on? The results should consider all the variables you collected.

Discussion. The results are overall poorly discussed. I suggest to follow  STROBE guidelines for cross sectional studies, with those elements:

  • Summarize key results with reference to study objectives
  • Discuss limitations of the study, taking into account sources of potential bias or imprecision.
  • Give a cautious overall interpretation of results considering objectives, limitations, multiplicity of analyses, results from similar studies, and other relevant evidence.
  • Discuss the generalizability (external validity) of the study results

All these points are strongly lacking in your discussion.  In particular, your findings have not been compared with the field literature. This aspect strongly limits the generalizability of your findings.

Conclusion. The novelty of your findings are questionable. You should also carefully explain what is the specific contribution that your findings bring to literature and knowledge in this area. Please be very clear on what your study adds, exactly how it extends previous knowledge. What emerges is that increasing workload has a detrimental effect on perceived stress. This is not new. Several relevant factors are omitted in the analysis, strongly limiting the relevance of the findings.

Best Regards

Author Response

Dear Reviewer 2,

Thank you very much for your very useful comments to qualify our manuscript better with scientifically academic requirement. 

Please see attachment file is our response to your comments

Reviewer 3 Report

This report documented HCW's feelings/experiences during a wave of the COVID19 pandemic. The design itself was very simple, and one used by a variety of researchers in a number of studies. The fact that I think the manuscript is important/noteworthy is solely because it offers information about HCW's in unprecedented  times - a pandemic to a novel virus. 

There was, in my opinion, only one thing that was surprising about the results, and that was the number of HCW's involved in direct coronavirus contact reporting stress/significant stress was so low - I find under 50% amazing given the workload and all of the unknowns about the virus. Good for them!

Author Response

We agree a 44% prevalence of stress is not too high. Nevertheless, it is important to remove all potential stress factors in order to avoid mental problems among HCW and improve their work performance. 

Reviewer 4 Report

Very interesting paper.

Author Response

Thanks for your kind appreciation of our manuscript

Round 2

Reviewer 1 Report

Dear Authors,

I respect that you do not agree. But there is a lot of scientific literature that looks at stress in each category. Claiming that a lab technician can have the same stress as a doctor seems inappropriate to me. The turnover, the type of work, or the contact with patients is totally different. In fact, a laboratory technician has no contact with patients. Therefore, the stress generated by the fear of being infected does not exist, for example. I am sorry that you do not want to carry out your analyzes as I have suggested. Therefore I must reject your manuscript.

Best regards

Author Response

Dear Reviewer 1,

With regard with your comment, we appreciated your good point in asking for stress distribution specified in different occupational groups is needed and more convincing explanation. We add descriptive characteristics specifically for the different occupations and explain these results in the text (see Table 1). We also update related references of occupation-specific stress in our Discussion section to support laboratory technician’s also at high risk of stress experience during the pandemic.  

It’s due to convenient sampling technique and small sample size, we could not include each occupational group in our multiple logistic regressions, but we classify them into two categories based on criteria of direct and indirect contact with patients and bio-samples (Table 3). We also understand that stress varies across different occupational groups, but in this fight against COVID-19, every frontline healthcare worker is at risk for stress, regardless of whether it is a physician or laboratory worker (see our further explanation in Discussion section as well). Attached file is our revised manuscript upon your recommendation

Thanks again for your very useful comments to improve our manuscript for considering publication.

Reviewer 2 Report

Dear Authors

I carefully evaluated the revised version of your paper, finding it substantially improved with respect to the first version.

All my concerns have been properly addressed

Best Regards

Author Response

Dear Reviewer 2,

Thanks for your useful comment that it helps us to improve better our manuscript for publication consideration 
